# Quantifying the Variability of Phytoplankton Blooms in the NW Mediterranean Sea with the Robust Satellite Techniques (RST)

**Emanuele Ciancia** [1,2,*], **Teodosio Lacava** [1], **Nicola Pergola** [1], **Vincenzo Vellucci** [3], **David Antoine** [4,5], **Valeria Satriano** [2] and **Valerio Tramutoli** [2]

[1] Institute of Methodologies for Environmental Analysis, Italian National Research Council, C.da Santa Loja, Tito Scalo, 85050 Potenza, Italy; teodosio.lacava@imaa.cnr.it (T.L.); nicola.pergola@imaa.cnr.it (N.P.)

[2] School of Engineering, University of Basilicata, Via dell'AteneoLucano 10, 85100 Potenza, Italy; valeria.satriano@unibas.it (V.S.); valerio.tramutoli@unibas.it (V.T.)

[3] Institut de la Mer de Villefranche, CNRS, Sorbonne Université, LOV, F-06230 Villefranche-sur-Mer, France; enzo@imev-mer.fr

[4] Remote Sensing and Satellite Research Group, School of Earth and Planetary Sciences, Curtin University, Perth, WA 6845, Australia; david.antoine@curtin.edu.au

[5] Laboratoired'Océanographie de Villefranche, CNRS, Sorbonne Université, IMEV, F-06230 Villefranche-sur-Mer, France

[*] Correspondence: emanuele.ciancia@imaa.cnr.it; Tel.: +39-0971-427242; Fax: +39-0971-427271

**Abstract:** Investigating the variability of phytoplankton phenology plays a key role in regions characterized by cyclonic circulation regimes or convective events, like the north-western Mediterranean Sea (NWM). The main goal of this study is to assess the potential of the robust satellite techniques (RST) in identifying anomalous phytoplankton blooms in the NWM by using 9 years (2008–2017) of multi-sensor chlorophyll-a (chl-a) products from the CMEMS and OC-CCI datasets. Further application of the RST approach on a corresponding time-series of in situ chl-a measurements acquired at the BOUSSOLE site allows evaluation ofthe accuracy of the satellite-based change detection indices and selecting the best indicator. The OC-CCI derived chl-a anomaly index shows the best performances when compared to in situ data ($R^2$ and RMSE of 0.75 and 0.48, respectively). Thus, it has been used to characterize an anomalous chl-a bloom that occurred in March 2012 at regional scale. Results show positive chl-a anomalies between the BOUSSOLE site and the Center of Convection Zone (CCZ) as a possible consequence of an intense convection episode that occurred in February 2012.

**Keywords:** phytoplankton phenology; multi-sensor ocean colour data; long-term analysis; North-Western Mediterranean Sea

## 1. Introduction

Variations in atmospheric and oceanic forcing (i.e., winds, intermittent upwelling, seasonal change in stratification, sea surface warming) and the effects of climate changes can impact some ecosystem properties, including marine primary production, the phytoplankton community structure and phytoplankton phenology [1–3]. Changes in phytoplankton phenology, such as timing and magnitude of the spring bloom, can produce harmful effects for the pelagic ecosystem, including mismatches with fish spawning, thus impacting fisheries [4–6]. In this scenario, characterizing seasonal and inter-annual variations of phytoplankton is a prerequisite for assessing how changes in the marine environment propagate from primary producers to higher trophic levels [7].

Satellite ocean colourradiometry (OCR) is a powerful tool to study phytoplankton phenology [8], as it provides synoptic and long-term observations of the sea-surface chlorophyll-a concentration (chl-a), a proxy of phytoplankton biomass [9]. Several studies have been conducted to investigate chl-a seasonal patterns at basin/regional scales by using multivariate clustering methods with OCR data [10–13].

Although the chl-a seasonality is well documented at these scales of analysis, the assessment of its inter-annual variability deserves to be better investigated [14]. To this aim, Barale et al. [15] analysed the chl-a inter-annual variability in the Mediterranean Sea by using multi-annual (1998–2003) Sea-viewing Wide Field-of-view Sensor (SeaWiFS) data, thus detecting chl-a anomalies in some specific areas such as the Gulf of Lion and the Rhodes gyre. Based on a merged (SeaWiFS and Moderate Resolution Imaging Spectrometer (MODIS)) dataset (1998–2014), Mayot et al. [16] investigated the chl-a inter-annual variability over well-identified bio-regions [10] highlighting the episodic occurrence of new trophic regimes, especially in the north-western Mediterranean Sea (NWM). Other studies based on OCR time series revealed how irregular (i.e., non-seasonal) inter-annual chl-a variations can be found in areas affected by intense wind forcing or winter deep water convection (WDWC) events such as the NWM, the Alboran Sea and the Adriatic Sea [7,14,17,18].

NWM can be considered a test area to study the interplay between phytoplankton bloom dynamics and the WDWC events as they represent the major processes driving nutrient availability into the euphotic layer and thus the open-ocean primary production [19,20]. Although WDWC is an annual feature in the NWM, its inter-annual fluctuations determine a high variability in terms of magnitude and spatial extension of the spring bloom [21–24]. In this context, it is worth implementing tools capable of identifying the non-seasonal (i.e., irregular) chl-a variations [25,26].

The robust satellite technique (RST) [27] is a general change detection scheme based on the analysis of long-term datasets homogeneous in the spatiotemporal domain which was already applied for studying sea-surface processes with satellite imagery [28–30]. The inherent rationale for the RST approach is to discard any cyclical fluctuations (daily or seasonal) of a geophysical variable in order to identify only its statistically infrequent variations and therefore are defined as anomalies (Tramutoli et al., 2007). Ciancia et al. [30] successfully applied the RST approach to identify anomalous phytoplankton blooms in the Gulf of Taranto (north-western Ionian Sea) by using 12 years (2003–2015) of MODIS-Aqua Level 3/Level 2 chl-a data. However, the long-life sensor issues of MODIS (i.e., degradation in the sensor detector response) could introduce noise in the derived products [9,31] and so the application of the same method on the MODIS archive will not be feasible on the long-term.

The recent availability of homogenized and inter-calibrated time series of multi-sensor (SeaWiFS, Medium Resolution Imaging Spectrometer (MERIS), MODIS-Aqua, Visible Infrared Imaging Radiometer Suite (VIIRS)) OCR products provided by the European Space Agency (ESA) Ocean Colour Climate Change Initiative Program (OC-CCI) and the European Union (EU) Copernicus Marine Environmental Monitoring Service (CMEMS) should enablethe aforementioned limitations to be overcome [26,32,33] for the RST implementation. Furthermore, their exploitation increases the probability of valid clear-sky observations and the derived chl-a products easily adapt to different bio-optical conditions being based on blended or switchable chl-a algorithms [32,33]. All these factors can contribute toreducing the potential sources of noise in the historical signal measured thus improving the RST sensitivity in detecting slight changes and subtle anomalies.

This paper aims at testing the RST approach on a new study area (i.e., NWM) by multi-sensor merged data, thus requiring a preliminary assessment through sea-truth data. The in situ measurements acquired by a permanent optical mooring, namely Bouée pour l'acquisitiond'unesérieoptique à long terme (BOUSSOLE), located in the Ligurian Sea, can be profitably exploited for this purpose as it provides collections of quasi-continuous data since 2003 [34].

Our threefold objective is (1) to evaluate the potential of the RST approach on a time series (2008–2017) of in situ chl-a measurements; (2) to validate results of the RST methodology applied to 9 years (2008–2017) of multi-sensor merged chl-a products from the CMEMS and OC-CCI (version 4.2) against corresponding in situ data; (3) extend the single location analysis to a regional scale (i.e., NWM) in order to characterize the chl-a anomalies in the spatio-temporal domain.

## 2. Materials and Methods

### 2.1. Study Area

The NWM (Figure 1) is one of the most productive areas of the Mediterranean Sea, recording annual primary production values ranging between 86 and 232 gC m$^{-2}$ [35]. The NWM is characterized by meso-to oligotrophic conditions varying from spring to summer depending on the seasonal fluctuations of the physical forcing (i.e., winter mixing and thermal stratification) [35]. In particular, the area shows characteristics of a typical temperate region, with a phytoplankton bloom occurring in late winter-early spring months and it is classified as a "blooming" bio-region [10,16,36,37].

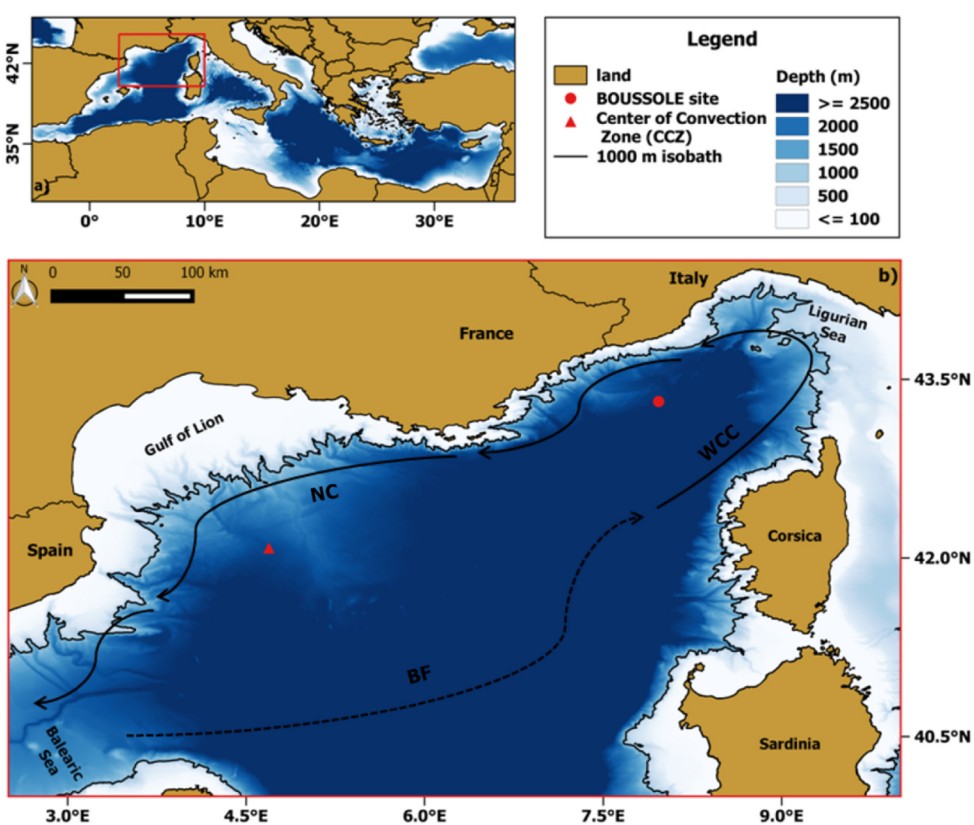

**Figure 1.** (**a**) Localization (highlighted in red) of the NWM within the Mediterranean basin; (**b**) magnification of the study area within the red box of (**a**). Two sites of interest are represented by a red circle (BOUSSOLE site) and triangle (center of convection zone (CCZ)), respectively. The mean currents are depicted by arrows with continuous black lines (WCC: western Corsican current, NC: northern current) while the Balearic front (BF) by an arrow with a dashed black line.

The blooming area roughly coincides with the large-scale cyclonic circulation, delimited on the northwest by the northern current (NC) flowing southward along the coastline, on the south by the permanent Balearic front (BF), and by the western Corsicancurrent (WCC) [23,38], as showed in Figure 1b. In addition, the variability in time, space and magnitude of phytoplankton blooms is also due to the physical processes characterizing the NWM [39]. For instance, the NWM is known as a region where WDWC events take place, with a center located at about 43.0°N, 4.7°E (red triangle in Figure 1b) [40]. The cold and dry local winds (i.e., Mistral and Tramontana) blowing on the NWM can erode the near-surface stratification thus inducing WDWC, that impacts the upper ocean through the supply of nutrients necessary to feed the successive spring bloom [24,39].

The BOUSSOLE site (43.37°N, 7.9°E) is located in the Ligurian Sea, in one of the NWM sub-basins, at about 32 nautical miles from the French coast (water depth is 2440 m) [41]. In particular, BOUSSOLE is in the central area of the cyclonic circulation characterizing the Ligurian Sea where the prevailing ocean currents were usually weak (<20 cm/s) within

most of the deployment years (18 years) [34]. The low-current pattern of the site together with the distance from shore are crucial characteristics making the site a satisfactory location to calibrate and validate OCR observations [41].

The BOUSSOLE site shows a marked seasonality of the physical conditions [42] switching from deep (~400 m depth) mixed layers in winter to a prevailing stratification in summer (~20 m) [43,44]. It displays generally oligotrophic conditions especially in summer with chl-a values <0.1 mgm$^{-3}$ (minima ~0.05 mg/m$^{-3}$) and undetectable nitrate levels. The early spring bloom period (i.e., from February to March-April) usually produces an increase in chl-a up to 3–5 mgm$^{-3}$ because of the nitrate enriched waters [41].

### 2.2. In Situ Chl-A Data

The BOUSSOLE site is visited monthly since July 2001 to acquire 0–400-m casts of hydrological (conductivity–temperature–depth, CTD) data, complementary inherent optical properties (IOPs) and apparent optical properties (AOPs), water samples for subsequent phytoplankton pigment analyses (high-performance liquid chromatography, HPLC) and particulate absorption measurements [43,45].

In addition, the deployment of two sets of WetLabs (now Sea-Bird Scientific) ECOFLN-TUs fluorometers (470 nm excitation, 695 nm emission) on the optical mooring (at 4 and 9 m depths, respectively) has allowed for the acquisition of chl-a fluorescence (Fluo, relative units) since 2007 [14]. Copper shutters on instrument optics are used to minimize biofouling. Sixty-second measurement sequences are recorded every 15 min and the median value of each sequence is kept as representative of the measure.

For each day, average Fluo data acquired before dawn, Fluo1, and after sunset, Fluo2, are used to filter out data potentially affected by non-photochemical quenching [14]. The average value of Fluo1 and Fluo2 is then used as daily Fluo value, i.e., approximately the interpolated value at solar noon (about 11h30′ UTC local time). The daily Fluo is converted into daily chl-a (mg m$^{-3}$) by the following relationships based on log-linear regression analysis of fluorescence and total chl-a from HPLC analyses at 5 and 10 m depth:

$$\ln[chl-a_{z_1}] = 0.8183 \ln[Fluo_{z_1}] - 0.171 \qquad (1)$$

$$\ln[chl-a_{z_2}] = 0.7182 \ln[Fluo_{z_2}] - 0.2484 \qquad (2)$$

where $z_1$ and $z_2$ are the shallowest and deepest depth, respectively. Finally, the optically weighted chl-a value is approximated as:

$$chl-a = \frac{2[chl-a]_{z1} + [chl-a]_{z2}}{3} \qquad (3)$$

Within the purpose of this work, we selected the 2008–2017 period to implement the RST approach.

### 2.3. Satellite Chl-A Data

Both the OC-CCI and CMEMS dataset are mostly based on a common processing chain and include daily chl-a products developed to be used for long-term studies (since 1998, ongoing). However, they differ by the spatial resolution, the optical classification schemes and the algorithms for chl-a retrievals they are based on.

Focusing on the OC-CCI data, we considered the Level 3 daily chl-a product at 4 km spatial resolution [46], hereafter OC-CCI chl-a. This is a merged product derived from the OC-CCI remote-sensing reflectance, R$_{rs}$(λ), from SeaWiFS(1997–2010), MERIS (2002–2012), MODIS (2002-ongoing) and VIIRS (2011-ongoing). Inter-sensor bias is removed both by band-shifting [47] and bias-correcting the MODIS and MERIS R$_{rs}$(λ) values to the SeaWiFS reference values and merging is obtained through a weighted averaging procedure [7,48]. Then, the application of an optical classification scheme [49–51] allows for determining the best empirical OC-CCI chl-a algorithm, that is a result of a blended algorithm between OC3 [52], OCI [53] and OC5 [54]. In order to evaluate the membership percentage to each

water class at pixel level, we also extracted the daily "normalised water class membership" products with the corresponding daily chl-a.

The CMEMS data, considered here, is the Level 3 daily chl-a product at 1km resolution (OCEANCOLOUR_MED_CHL_L3_REP_OBSERVATIONS_009_073) [55] hereafter CMEMS chl-a. The CMEMS processing chain replicates the OC-CCI one (by usingRrs(λ) from SeaWiFS, MERIS, MODIS and VIIRS), except that all available daily pixels (originally at 4 km resolution) are remapped at 1 km resolution (on the equirectangular grid covering the Mediterranean Sea) before performing the band-shifting [32]. Then the procedure identifies two possible water types, *Case I* and *Case II*, following the optical classification scheme by D'Alimonte et al. [56] before deriving chl-a values with the Med-OC4 [57] or the AD4 [58] algorithms. The daily CMEMS chl-a product includes information of a water type mask at pixel level.

Both the OC-CCI and CMEMS chl-a products were downloaded for the same period of in situ chl-a measurements (2008–2017). It is worth noting that the satellite chl-a dataset used could be less populated than the in situ one because of filtered or flagged pixels (cloudy or no data).

### 2.4. The Robust Satellite Techniques

RST is a statistical method developed to analyse multi-annual series of satellite data [27] and has been already used for monitoring sea surface environmental phenomena [9,28–30]. RST requires the computation, at pixel level, of the climatological mean and standard deviation (called reference fields) of the investigated variable observed within homogenous conditions (i.e., same location, month and acquisition time). Although RST was developed for application on satellite data, here we also test its application on in situ chl-a data that meets the homogeneity requirement.

After calculation of reference fields, the Absolutely Local Index of Change of the Environment for chl-a, namely the chl-a ALICE index [30], is computed as:

$$\otimes_{chl-a}(x,y,t) = \frac{chl-a(x,y,t) - \mu_{chl-a}(x,y)}{\sigma_{chl-a}(x,y)} \tag{4}$$

where$chl-a(x,y,t)$ is the chl-a value at longitude $x$, latitude $y$ and time $t$, $\mu_{chl-a}(x,y)$ and $\sigma_{chl-a}(x,y)$ are the climatological monthly mean and standard deviation computed at the same location. These parameters are computed for the three 2008–2017 datasets considered here (i.e., CMEMS, OC-CCI and in situ), and we refer alternatively to each of them using the same symbols in the following.

A positive/negative chl-a ALICE value indicates an increase/decrease of chl-a with respect to its climatological value. For construction, the chl-a ALICE index is a standardized variable, which tends to have a Gaussian distribution. This is confirmed in Figure 2a–c where the frequency histograms of the chl-a ALICE index values are displayed with the theoretical Gaussian curve superimposed. The three chl-a ALICE indices follow a Gaussian distribution (with a μ~0 and a σ~1), although showing a slight positive skewness (withFisher–Pearsoncoefficients ofskewness ≤1). Therefore, the occurrence probability of values above/below ±1, ±2 and ±3 are about 16%, 2.5% and 0.13%, respectively.

We define a chl-a anomaly when the chl-a ALICE index is above/below ±2 (i.e., $|\otimes_{chl-a}(x,y,t)|>2$) representing a statistically significant level of occurrence probability.

### 2.5. Ancillary Data

In order to better characterize the chl-a anomalies at regional scale (Figure 1b), we used the CMEMS physical re-analysis MEDSEA_REANALYSIS_006_004 product. It is generated with a hydrodynamic model, (supplied by the Nucleus for European Modelling of the Ocean (NEMO)), and on a variational data assimilation scheme (OceanVAR) for temperature and salinity vertical profiles and satellite sea level anomaly along track data [59]. In particular, we downloaded and processed the mean daily fields of eastward

and northward sea water velocities (at about 4–5 km spatial resolution) to derive the mean daily velocity and direction of the sea surface current.

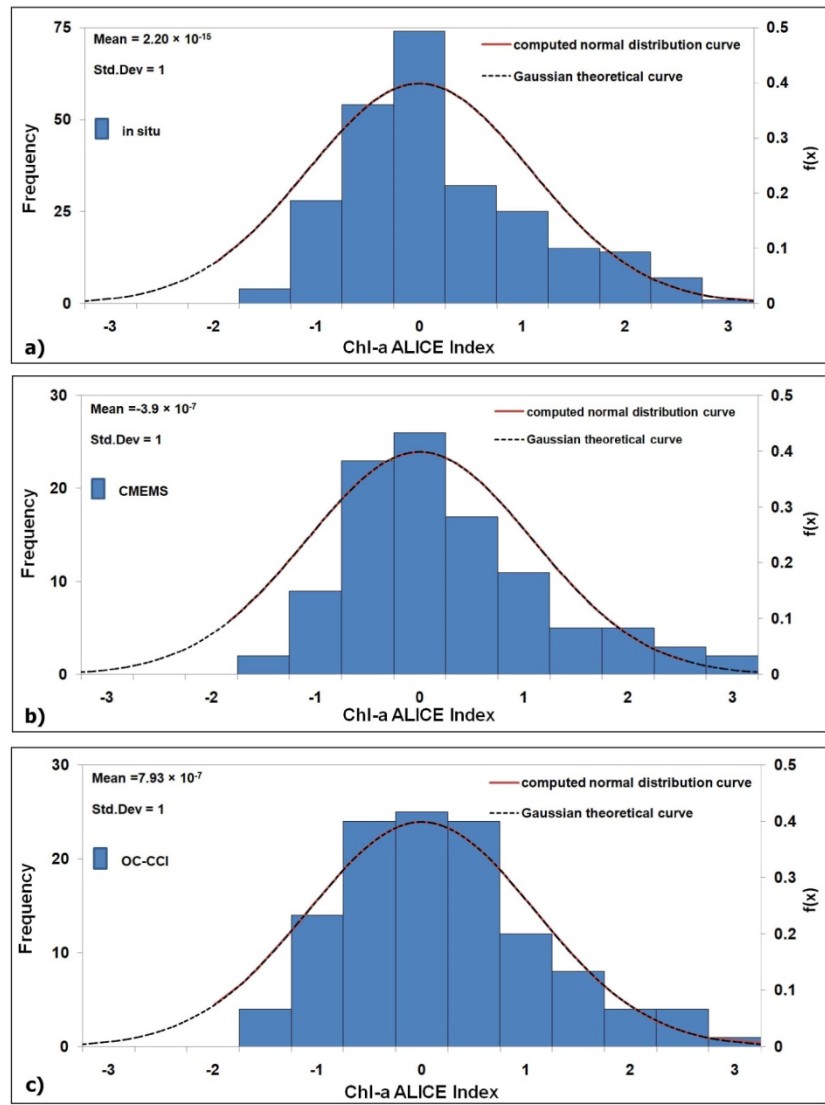

**Figure 2.** Frequency histograms of the chl-a Absolutely Local Index of Change of the Environment(ALICE) index computed at the BOUSSOLE site for the 2008–2017 December months with the computed and theoretical Gaussian probability density function, f(x)(red and dashed black lines, respectively), for: (**a**) in situ; (**b**) Copernicus Marine Environmental Monitoring Service (CMEMS, mean value of a 3 × 3 pixels box centred at the BOUSSOLE site); (**c**) Ocean Colour Climate Change Initiative Program (OC-CCI, pixel centred at the BOUSSOLE site).

Furthermore, we also analysed the sea surface heat losses to detect the related WDWC episodes and investigate their influence on the occurrence and timing of anomalous phytoplankton blooms. To this aim, we considered the ERA5 reanalysis dataset provided by the Climate Data Service (CDS) [60].Based on the European Centre for Medium-Range Weather Forecasts (ECMWF), ERA5 includes consistent atmosphere and sea surface analyses from 1979 to the present [61]. We downloaded the 3-hourly ERA5 data (at about 25 km spatial resolution) related to four components of the sea-air fluxes (i.e., surface net shortwave flux-$Q_{ssf}$, surface net thermal radiation-$Q_{str}$, surface latent heat flux-$Q_{slhf}$, surface sensible

heat flux-$Q_{sshf}$) from the CDS website and computed at pixel level the daily mean of each component to derive the daily Net Heat Flux ($Q_{net}$- Wm$^{-2}$) via the bulk formulae [62]:

$$Q_{net} = Q_{ssf} + Q_{str} + Q_{slhf} + Q_{sshf} \tag{5}$$

### 2.6. Match-Up Analysis

The performances of the RST approach on the NWM area by multi-sensor merged CMEMS and OC-CCI data need to be assessed through a comparison with corresponding in situ data. We first derived time series of OC-CCI chl-a data at the pixel centred on BOUSSOLE, and time series of CMEMS chl-a values for a 3 × 3 box located at the BOUSSOLE site to homogenize their spatial resolutions. For the OC-CCI dataset, all the flagged pixels (cloudy or no data) were discarded as well those showing less than 50% membership to a given water class. For the CMEMS dataset, only the 3 × 3 boxes containing at least 50% of valid pixels (not flagged) and characterized by >50% (at least 5 pixels on the 3 × 3 box) of membership to one of the two water types were retained for further processing.

Match-up with in situ data were analysed with the following statistical indicators, such as the coefficient of determination ($R^2$), the $p$-value, and the root mean square error (RMSE), defined as:

$$MSE = \sqrt{\frac{1}{N} \sum_{i=1}^{N} (x_i - y_i)^2} \tag{6}$$

wherein $x_i$ is the $i_{th}$ satellite-derived chl-a ALICE value (OC-CCI/CMEMS), $y_i$ is the $i_{th}$ in situ-derived one, and N is the number of match-up.

## 3. Results

### 3.1. Climatological Analysis

One of the main goals of this work is to evaluate the RST suitability to detect anomalous phytoplankton blooms at the BOUSSOLE site and within the NWM area. For this reason, we performed a preliminary analysis of the climatological chl-a cycle to identify the bloom timing and the months where focusing the RST analysis. To this end, we computed the 8-day composite climatology of in situ chl-a data at the BOUSSOLE site (Figure 3).

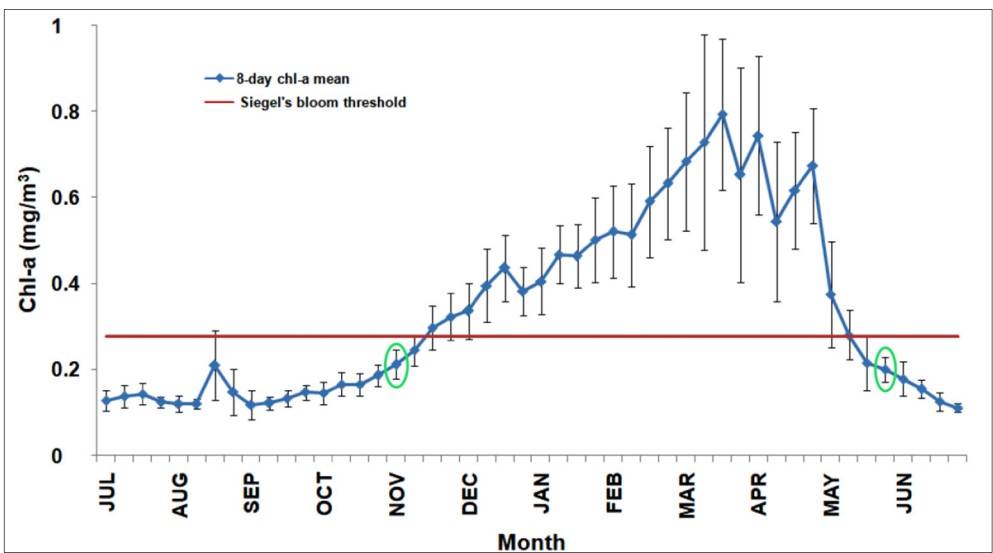

**Figure 3.** Eight-day chl-a climatology at the BOUSSOLE site for 2008–2017 (blue line) with standard deviation(black vertical bars).The red line indicates the chl-a threshold defining the start/end of the bloom according to Siegel et al. [63]. The empty green circles define the bloom timing (start/end) according to Brody et al. [64].

This climatology follows the typical seasonal cycle of the NWM "blooming" trophic regime [10,36], as it displays a gradual chl-a increase in late winter/early spring reaching its maximum in spring (i.e., 0.8 mgm$^{-3}$ on the third week of March).

To define the bloom timing (start/end), we used the methods based on a chl-a threshold criterion proposed by Siegel et al. [63] and its revised version by Brody et al. [64]. The first method indicates a start of the bloom the third week of November and its end the first week of May, respectively. The Brody et al. [64] approach puts the start of the bloom at the first week of November and its end at the third week of May when chl-a values are below the previous threshold (red line in Figure 3) for two consecutive weeks. The second method is more robust as it reduces the transient effects of data noise [64], and is used here as a reference to select the months where focusing the RST analysis (i.e., November–May 2008–2009, November–May 2009–2010, . . . etc.).

### 3.2. Validation of the Satellite Chl-A Absolutely Local Index of Change of the Environment(ALICE)

Figure 4 shows the temporal variability of the three chl-a ALICE indices considered here for each November–May period of the 2008–2017 time series. Although some of the satellite time series show data gaps due to flagged pixels, they display a general good agreement with in situ time series for most of the considered periods. In particular, all the three chl-a ALICE indices are able to concurrently detect statistically significant anomalies (i.e., chl-a ALICE Index > 2) at the BOUSSOLE site, as for example in March 2012, May 2012 and May 2016.

To assess the accuracy of the satellite-based chl-a ALICE indices we performed a match-up analysis with the corresponding in situ values. For consistency, we re-sampled the in situ chl-a dataset to satellite data frequency. Figure 5a,b show the scatter plots of the match up for the CMEMS and OC-CCI datasets, respectively.

The accuracy of the two satellite-derived chl-a ALICE indices were evaluated by the regression indices and statistical indicators summarized in Table 1. The chl-a ALICE OC-CCI exhibits better performances with a higher $R^2$ value (0.75) and a lower RMSE value (0.48) as compared to the chl-a ALICE CMEMS. This result is probably due to the higher accuracy of the blended algorithm the OC-CCI chl-a products are based on. An evaluation of the CMEMS and OC-CCI chl-a algorithms can be found in Appendix A.

### 3.3. Regional Scale Analysis: The March 2012 Case Study

The better performances of the chl-a ALICE OC-CCIsuggested its exploitation for further insights at regional scale. As shown in Figure 4d, the chl-a ALICE OC-CCIrecorded statistically significant anomalies (i.e.,chl-a ALICE values >2) in March 2012 at the BOUSSOLE site, indicating the occurrence of a phytoplankton bloom deserving to be in-depth investigated. The variability of the OC-CCI derived ALICE values (with the corresponding in situ ones) in March 2012 is shown in Figure 6.

The two chl-a ALICE indices show a good agreement (with a $R^2$ value of 0.9 and *p*-value < 0.001) thus inherently proving the effectiveness of the RST approach in detecting relative variations of the investigated signal regardless the different source of data (satellite vs. in situ). Both the chl-a ALICE indices identify persistent chl-a anomalies at the BOUSSOLE site on 6 days (i.e., 16, 19–22, 24 March 2012).

To characterize the spatial extension of the chl-a anomalies we computed the daily OC-CCI chl-a ALICE maps over the whole NWM area. Figure 7 shows the days with less than 50% of cloudy (or flagged) pixels within the 16–24 March 2012 period (i.e., 16, 19, 22, 24 March 2012) and includes also two days (i.e., 27, 30 March 2012) after the bloom event to better characterize the dynamics of chl-a anomalies.

The occurrence of chl-a anomalies mostly characterized the first four days (i.e., 16, 19, 22, 24 March 2012), especially on 16 March 2012, with the highest percentage of chl-a ALICE values > 2 (about 26.5% of the valid pixels in the scene). Although some chl-a anomalies are close to the Western Corsica coastline, the area with higher chl-a ALICE values was the region between the BOUSSOLE site and the CCZ. In particular, the anomalous chl-a pattern

was persistent around the BOUSSOLE site during the 16–24 March period, probably due to the cyclonic circulation regime characterizing the area, as shown by the sea surface current data (Figure 7a,d).

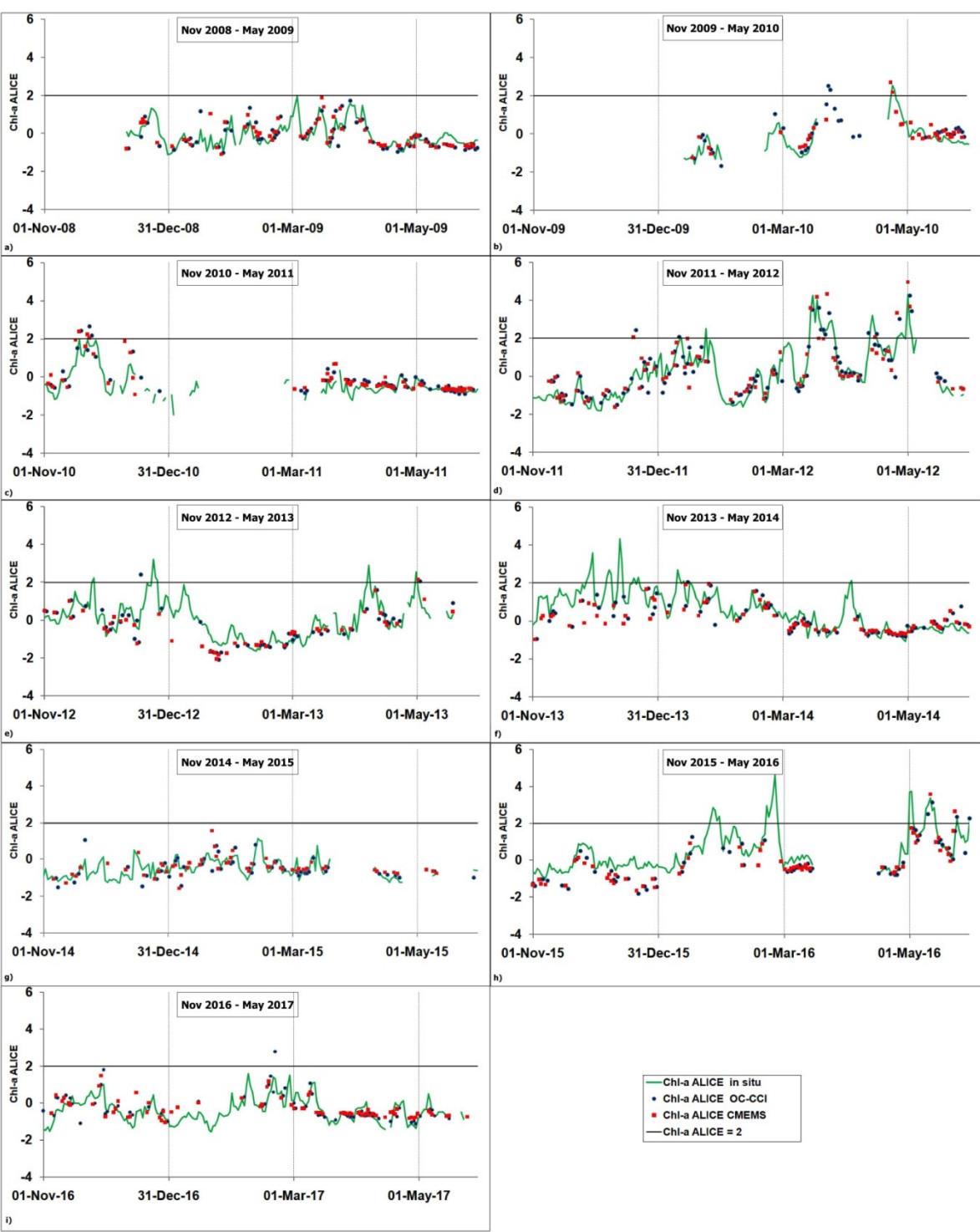

**Figure 4.** Temporal variability of the three chl-a ALICE indices, namely chl-a ALICE CMEMS (red squares), chl-a ALICE OC-CCI (blue circles) and chl-a ALICE in situ (green line). The (**a**–**i**) plots refer to each November–May period of the 2008–2017 investigated time series. The solid black line indicates a chl-a ALICE =2.

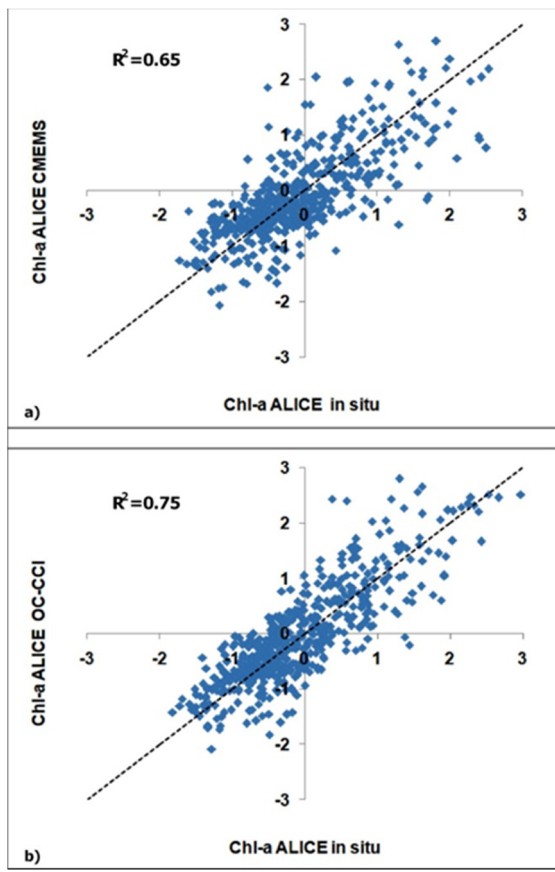

**Figure 5.** Scatter plots of the satellite-derived chl-a ALICE indices versus the corresponding in situ one, where (**a**) refers to the CMEMS dataset and (**b**) to the OC-CCI dataset.

**Table 1.** Regression indices and statistical indicators for thematch-up analyses of the satellite-derived chl-a ALICE indices.

| chl-a ALICE | N | R2 | *p*-Value | RMSE |
|---|---|---|---|---|
| Copernicus Marine Environmental Monitoring Service (CMEMS) | 550 | 0.65 | <0.001 | 0.57 |
| Ocean Colour Climate Change Initiative Program (OC-CCI) | 588 | 0.75 | <0.001 | 0.48 |

The chl-a ALICE values decreased in the BOUSSOLE area after the 24 March 2012 as the chl-a anomalous features clearly moved southward along the NC main flow. For this reason, the chl-a concentration returned to usual values for the seasonal period in most of the area except for the south-western zone where residual chl-a anomalies still occurred.

Based on this evidence, we can hypothesize that a strong deep-water convection event together with the permanent large-scale cyclonic circulation of NWM might have induced the occurrence of such an anomalous phytoplankton bloom.

*3.4. Influence of Winter Deep Water Convection (WDWC)Event on the March 2012 Anomalous Chl-A Bloom*

The achieved results lead to assume the occurrence of a WDWC event before the NWM phytoplankton bloom of mid-March 2012. In this regard, several authors have considered winter 2012 (i.e., from December 2011 to February 2012) as exceptional over the North Atlantic and European region [65] and the strongest in terms of heat losses within the 2008–2012 winter period [40]. In this context, we looked at the daily Qnet values at the

BOUSSOLE site and in the CCZ (Figure 8) to detect episodes of intense heat losses at the sea surface and associated WDWC events that potentially occurred during the January–March 2012 period. The two sites show a strong episode of net heat loss during the first half of February 2012 (10–12 days) followed by a less intense short-term event in early March 2012 (~3 days), after the water column stratification (i.e., the first day of the year when the Qnet becomes positive).

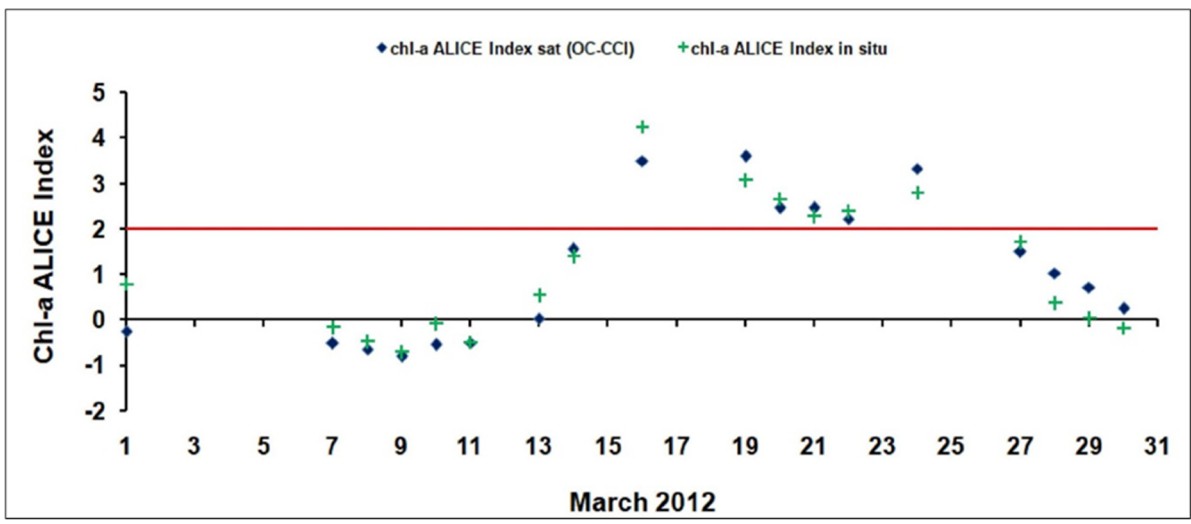

**Figure 6.** The temporal variability of the OC-CCI (blue rhombuses) and in situ (green crosses) ALICE indices at the BOUSSOLE site in March 2012. The solid red line indicates a chl-a ALICE =2.

The 03–10 February 2012 period displays the highest averaged Qnet values of about $-543$ $Wm^{-2}$ and $-688$ $Wm^{-2}$ for the BOUSSOLE and CCZ sites, respectively. The magnitude of such a net heat loss could indicate a WDWC event that affected the anomalous phytoplankton bloom in March 2012. To address this assumption, we performed a cross-correlation analysis between the Qnet values and the corresponding Chl-a ALICE ones within the January–March 2012 period. To provide a statistically significant analysis, we considered only the in situ Chl-a ALICE values by exploiting their higher availability than the corresponding satellite-derived ones. Figure 9 shows the cross-correlation plot between Qnet and in situ Chl-a ALICE at the BOUSSOLE site for the 90-day (January 2012–March 2012) time of analysis.

The relatively high correlation at 0 lagindicates that positive chl-a anomalies are concurrent with net heat increases and vice versa. It means that the phytoplankton bloom in March 2012 occurred with water column stratification (i.e., positive Qnet values), as shown in Figure 8a. The highest negative correlation is recorded at negative lags (i.e., $-0.52$ at $-44$ days) thus suggesting that the positive chl-a anomalies of mid-March 2012 are associated with intensive net heat losses at the beginning of February 2012. For this reason, we derived the corresponding OC-CCI 8-day chl-a map for the NWM area to delineate the spatial extent of the WDWC zone. To this aim, we adopted a threshold chl-a value as an index of the active vertical mixing of water column [42,65]. Considering that this value ranges between 0.15–0.25 $mgm^{-3}$ for the study area [40], we used here the threshold of 0.25 $mgm^{-3}$ given the slight overestimation of the OC-CCI algorithm for chl-a values <0.15 $mgm^{-3}$(see Figure A1b in Appendix A).

Figure 10a shows the extent of the WDWC zone by applying the threshold chl-a value to the 03–10 February 2012 composite chl-a map. Then we compared the WDWC extension of February 2012 with the chl-a anomalies of 16 March 2012 (day with highest occurrence of values >2), in order to identify a potential cause–effect relationship between the two (Figure 10b).

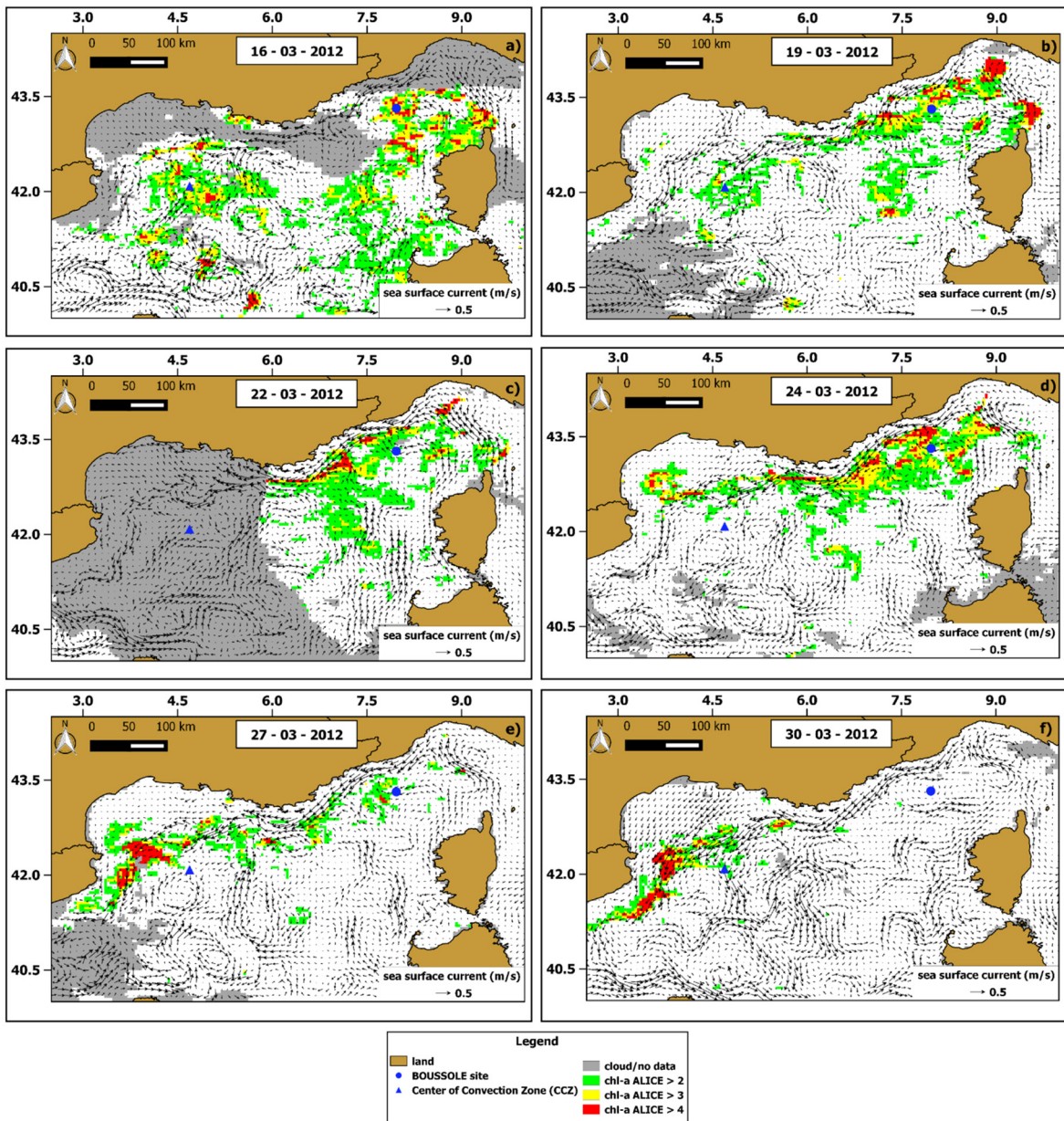

**Figure 7.** Chl-a ALICE OC-CCI maps of 6 days between 16–30 March 2012. The maps (**a**–**f**) refer to each day considered, namely 16, 19, 22, 24, 27 and 30 of March 2012, respectively. The three different colours indicate increasing (from green to red) statistical significance levels of the chl-a ALICE OC-CCI. The superimposed black arrows are the corresponding daily data of sea surface current (speed and direction).

The WDWC zone encompasses most of the open-ocean except for the shelf area within the Gulf of Lion as well as part of areas close to Corsica and Sardinia. Most of the chl-a anomalies detected on 16 March 2012 fall into the WDWC zone with a percentage of about 78%, thus apparently indicating a possible link between the strong WDWC episode of February 2012 and the anomalous chl-a bloom of mid-March 2012.

Finally, we can conclude that the area most affected by the February 2012 WDWC event roughly coincides with the large cyclonic gyre of the NWM area. Although such a permanent large-scale circulation is probably the primary cause for the phytoplankton bloom extension [42], the anomalous behaviour of March 2012 evidenced by the RST approach can be ascribed to the intense convection episode of February 2012 as described by several studies [40,42,65].

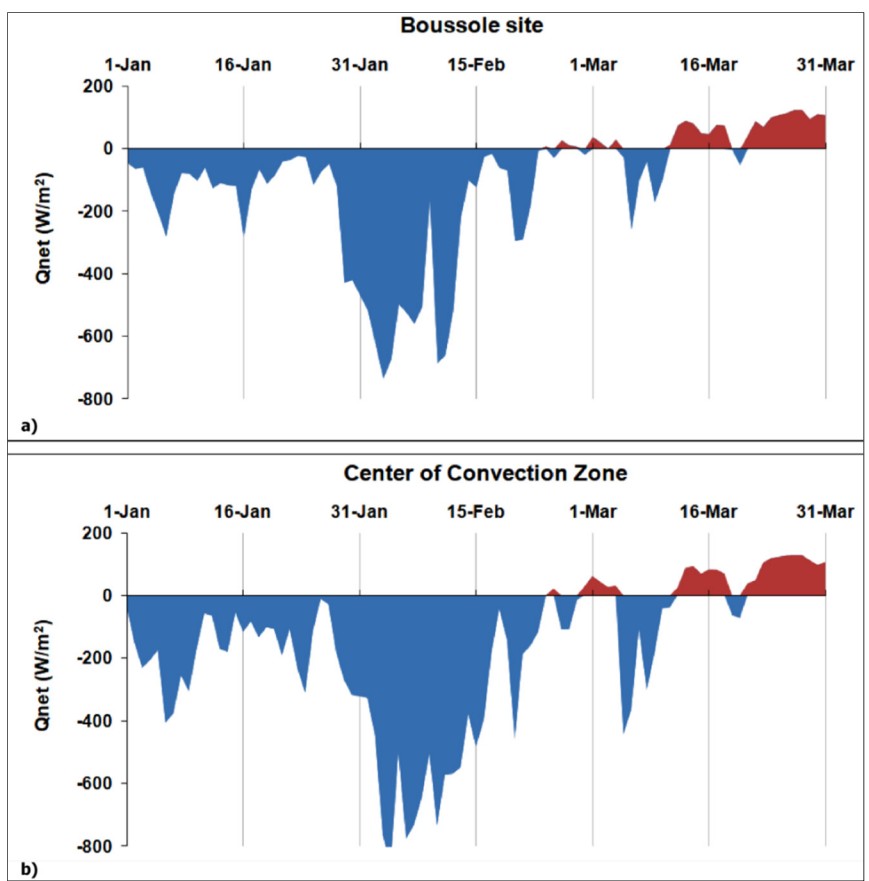

**Figure 8.** Temporal variability of the daily net heat fluxes calculated within the January–March 2012 period of analysis. (**a**) BOUSSOLE site and (**b**) Center of Convection Zone (CCZ).

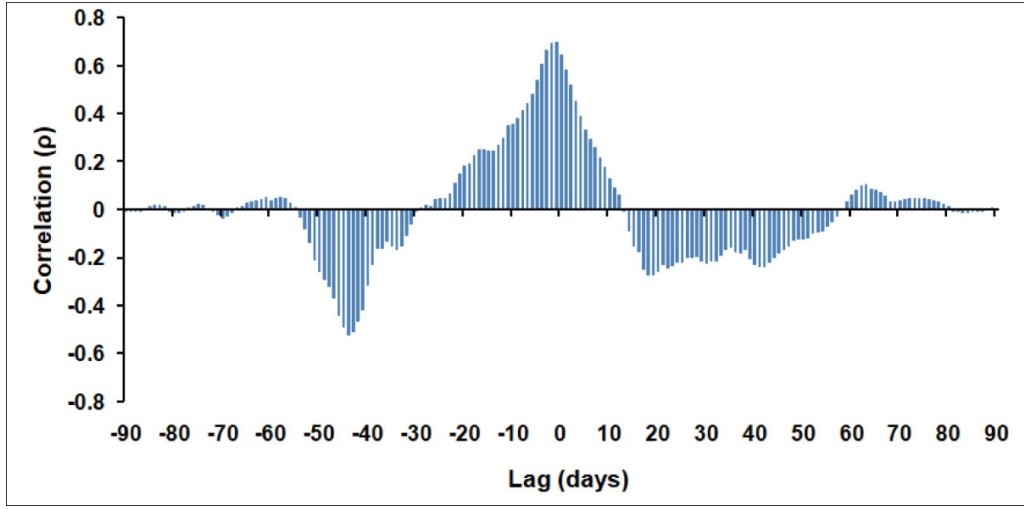

**Figure 9.** Cross-correlation (Spearman's ρ) between Qnet and in situ Chl-a ALICE values at the BOUSSOLE site within the January–March 2012 period of analysis.

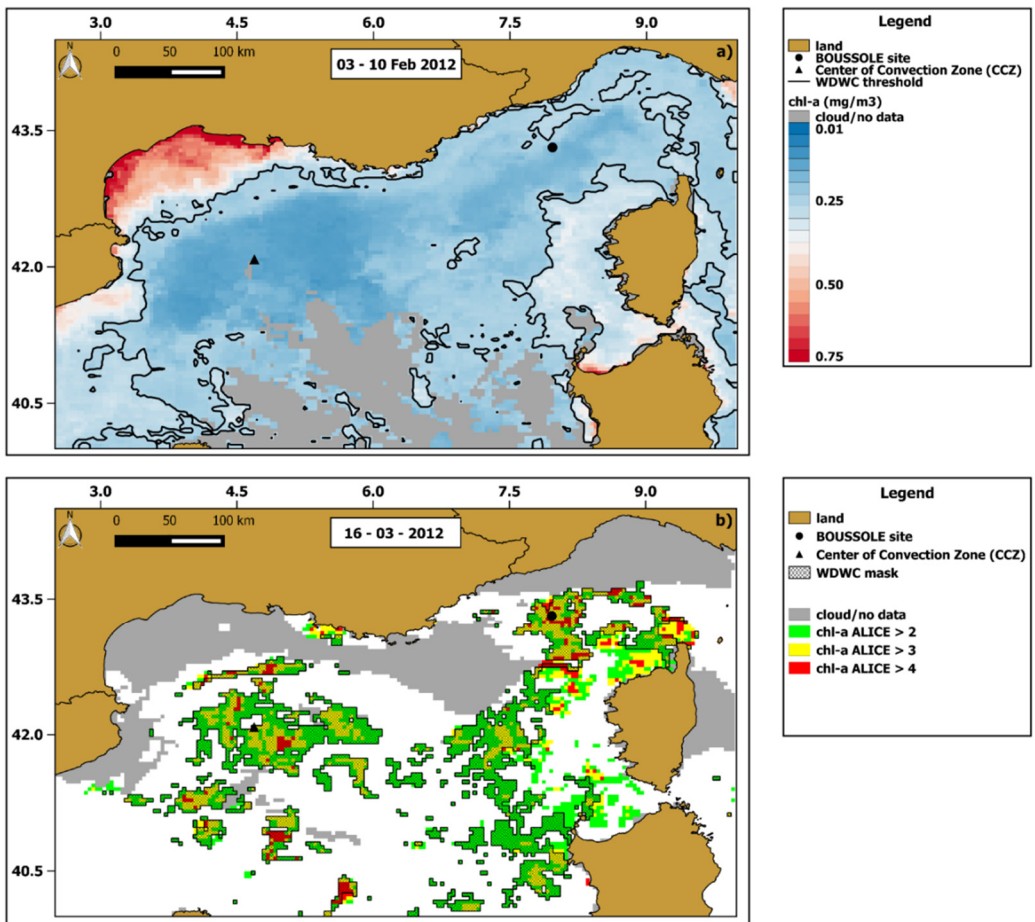

**Figure 10.** (**a**) The 8-day (03–10 February 2012) chl-a map where the area enclosed by the black solid line represents the winter deep water convection (WDWC) zone. (**b**) chl -a ALICE OC-CCI map of 16 March 2012. The masked pixels indicate the anomalous ones (chl-a ALICE ≥ 2) falling into the previously delimited WDWC area.

## 4. Conclusions

The phytoplankton phenology is an invaluable indicator for assessing how marine ecosystems react to external forcing [37,66]. Although the seasonality represents the most relevant factor influencing the chl-a variability at the Mediterranean Sea scale [17,18], the non-seasonal (irregular) component plays a crucial role in some regions characterized by cyclonic circulation regimes and blooming dynamics such as the Alboran Sea, the Adriatic Sea and the NWM [7]. In these areas phytoplankton phenology appears to be closely related to regional scale processes usually associated with convective events caused by winter cooling and strong winds [67,68]. In this context it is worth developing metrics to describe phytoplankton phenology and detect local-induced chl-a variations due to environmental forcing.

The exploitation of long-term data based on multi-sensor merged observations allowed developing and implementing ocean monitoring indicators (OMIs) able to track the relative changes of the essential climate variables (ECVs—recognised by the Global Climate Observing System (GCOS)), such as sea surface temperature (SST) and chl-a concentration [25]. CMEMS makes the "regional annual chlorophyll anomaly" (RACA) indicator available for the Mediterranean Sea by using historical series (1997–2014) of multi-sensor merged OCR data [69]. RACA is computed by subtracting a reference climatology (1997–2014) from the annual chl-a mean on a pixel-by-pixel basis and has been mainly designed to investigate the effects of climate changes and their potential correlations with the North Atlantic Oscillation (NAO) or El Niño Southern Oscillation (ENSO) phenomena [13]. However, the annual scale of the analysis tends to smooth short-term (daily, weekly)

anomalies induced by regional scale processes thus hampering their identification. In fact, the inherent rationale of RACA is to identify if the chl-a annual mean is above/below the climatological annual mean without taking into account the site-specific natural variability (i.e., standard deviation). For this reason, the maximum value of the RACA in the Mediterranean Sea can assume a different meaning if recorded in a "non-blooming" area rather than in a "blooming" one characterized by a higher inter-annual chl-a variability.

The RST approach is designed to overcome this limitation as the chl-a ALICE index (Equation (1)), for construction, furnishes a measurement of the chl-a deviation from its normal value (i.e., climatological monthly mean), weighted for its natural variability [30]. The Gaussian distribution of the chl-a ALICE index provides a statistically-based assessment of locally-induced changes in chl-a concentration. Furthermore, the differential and self-adaptive nature of the RST methodology ensures its applicability to different water types or trophic regimes and thus its potential usability for the whole Mediterranean Sea.

In this work, we assessed the potential of the RST approach in identifying anomalous phytoplankton blooms in the NWM by using 9 years (2008–2017) of multi-sensor merged chl-a products from the CMEMS and OC-CCI datasets. The application of RST on a corresponding time series of in situ chl-a measurements at the BOUSSOLE site allowed us to evaluatethe accuracy of the satellite-based chl-a ALICE indices and selecting the best indicator as well. Although both CMEMS and OC-CCI chl-a ALICE indices demonstrated to be capable in detecting chl-a anomalies, the latter performed best when compared to in situ data ($R^2$and RMSE values of 0.75 and 0.48, respectively). This was probably due to the higher accuracy of the blended algorithm the OC-CCI chl-a products are based on, thus suggesting that a regional blended algorithm could be developed for the NWM to improve the accuracy of chl-a products.

Then we exploited the synoptic view of satellite data as well as the pixel-based and self-adaptive RST capabilities to move from a local to a regional scale analysis. The chl-a ALICE OC-CCIwas used to characterize the anomalous phytoplankton bloom of mid-March 2012. The area roughly encompassed between the BOUSSOLE site and the CCZ showed the highest persistence of positive chl-a anomalies (i.e., $\otimes_{chl-a}(x,y,t)$> 2), probably due to the cyclonic circulation regime characterizing the area.

Based on this analysis, it is reasonable to assume the influence of the WDWC episodes on the March 2012 anomalous phytoplankton bloom. The cross correlation analysis between the Qnet and in situ ALICEvalues at the BOUSSOLE site gave a statistical weight to such an assumption. The highest negative correlation (i.e.,−0.52) at −44 days suggests the positive chl-a anomalies of mid-March 2012 are associated tointensive net heat losses at the beginning of February 2012. In particular, the anomalous behaviour of March 2012 can probably be ascribed to the intense convection episode recorded in February 2012, considering that most of the chl-a anomalies detected on 16 March 2012 fall into the WDWC zone with a relevant percentage (78%). Within a future perspective there will be the need for acquiring additional data to better understand the relationships between physical forcing and the phytoplankton phenology. In particular, it would be worthanalysing time series of heat fluxes and mixed layer depths to better define the RST-based metrics and to in-depth investigate the phytoplankton phenology in the NWM.

**Author Contributions:** Conceptualization, E.C., V.V., N.P. and T.L.; Methodology, E.C., V.V., N.P., T.L. and V.T.; Validation, E.C. and V.V.; Data Curation, E.C., V.V. and V.S.; Writing–Original Draft Preparation, E.C. and V.V.; Writing–Review andEditing, E.C., V.V., N.P., T.L. and D.A.; Supervision, D.A. and V.T., Funding (BOUSSOLE project), D.A. and V.V. All authors have read and agreed to the published version of the manuscript.

**Funding:** The BOUSSOLE time series project is funded by the Centre National d'EtudesSpatiales (CNES) and the European Space Agency (ESA/ESRIN contract 4000119096/17/I-BG) and supported by the French Oceanographic Fleet for ship time.

**Acknowledgments:** MelekGolbol, Emilie Diamond, GrigorObolenski, Vincent Taillandier and Eduardo Soto Garçia were essential for work at sea during the BOUSSOLE monthly cruises. Celine Dimier, Josephine Ras and Mustapha Ouhssain from the SAPIGH analytical platform of the Laboratoired'Océanographie de Villefranche (CNRS-France) are warmly acknowledged for the analysis of pigments. We are grateful to Lawrence George III for editing on an earlier version of the manuscript.

**Conflicts of Interest:** The authors declare no conflict of interest.

## Appendix A

To understand the better performances of the chl-a ALICE OC-CCIas compared to chl-a ALICE CMEMS, we have to evaluate the corresponding chl-a data and their associated water classes. To this end, we performed a match-up analysis between the satellite (CMEMS/OC-CCI) chl-a values and the corresponding in situ measurements at the BOUSSOLE site. Figure A1a,b shows the derived scatter plots for the CMEMS OC-CCI chl-a values, respectively.

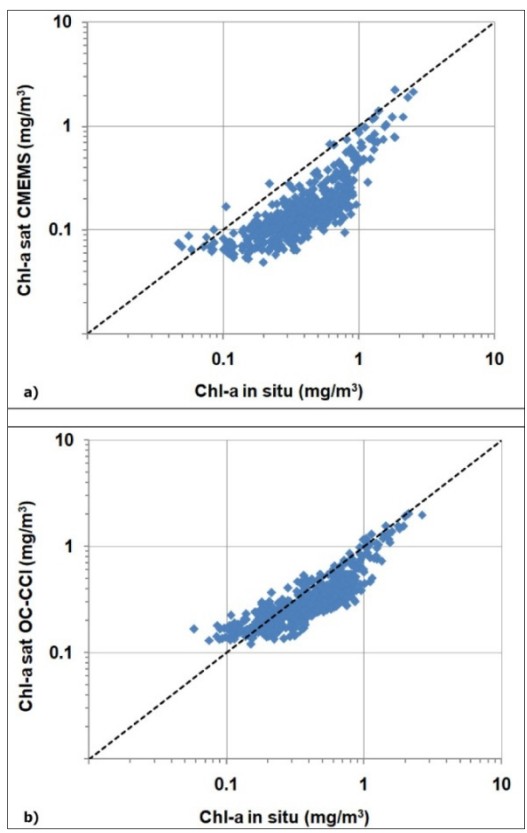

**Figure A1.** Scatter plots of the satellite-derived chl-a values versus the corresponding in situ one, where (**a**) refers to the CMEMS chl-a product and (**b**) to the OC-CCI chl-a one.

The accuracy of the two types of algorithms (i.e., switchable or blended for CMEMS and OC-CCI, respectively) were evaluated by the regression indices and statistical indicators summarized in Table A1. The OC-CCI blended algorithm exhibits a better accuracy than the CMEMS switchable algorithm, with a higher $R^2$value (0.80 vs. 0.66) and a lower RMSE value (0.18 vs. 0.33). Such a result could probably stem from different optical classification schemes the CMEMS and OC-CCI chl-a products are based on.

**Table A1.** Regression indices and statistical indicators for thematch-up analyses of the CMEMS/OC-CCI chl-a algorithms.

| Chl-A Algorithm | N | R2 | *p*-Value | RMSE (g/m$^3$) |
|---|---|---|---|---|
| CMEMS | 550 | 0.66 | <0.001 | 0.33 |
| OC-CCI | 588 | 0.80 | <0.001 | 0.18 |

We also counted the occurrence of *Case I/Case II* water types for the CMEMS dataset (at least 5 pixels on the $3 \times 3$ box) and computed the percentage membership to each of the 14 OC-CCI original classes for the OC-CCI dataset (pixel centred) as summarized in Table A2. For OC-CCI water classes, they have been reduced into three types: *open* water (classes1–7), *transitional* (8–12) and *coastal* (13–14) according to Jackson et al. (2017).

**Table A2.** Water type membership related to the BOUSSOLE site within the 7-month period considered (November–May).

| Dataset | Water Type | Membership (%) |
|---|---|---|
| **CMEMS** | *Case I* | 96.36 |
| | *Case II* | 3.64 |
| **OC-CCI** | *Open* | 12.76 |
| | *Transitional* | 86.89 |
| | *Coastal* | 0.35 |

The BOUSSOLE site falls into the *Case I* type for about 97% of the CMEMS dataset whereas it belongs to *transitional* water classes for about 87% of the OC-CCI dataset. This means that, for the CMEMS chl-a, theMedOC4 algorithm [57] has been used in 97% of the cases, whereas for the OC-CCI chl-a, the OC5 algorithm [54] was predominantly adopted within the blended rationale, being the optimal algorithm for transitional water type [51].

Although the OC-CCI classification scheme has low total classification scores in the Mediterranean Sea [51], these results suggest that its blended algorithm is more flexible than the switchable CMEMS algorithm and more capable to adapt to the peculiar optical characteristics of the NWM [57,70–73].

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
