# Peer review of "Quantifying the Variability of Phytoplankton Blooms in the NW Mediterranean Sea with the Robust Satellite Techniques (RST)"

_remotesensing, doi:10.3390/rs13245151_

Round 1

Reviewer 1 Report

A commented PDF is enclosed, where minor and major comments are included. 

Author Response

Please, see the attached file below

Reviewer 2 Report

Summary

This paper aims at testing the RST approach on a new study area (i.e. NWM) by multi-sensor merged data. They use this method to analyze the Absolutely Local Index of Change of the Environment for chl-a, namely chl-a ALICE index, and finally hypothesize that a strong deep-water convection event together with the permanent large-scale cyclonic circulation of NWM might have induced the occurrence of such an anomalous phytoplankton bloom. The structure is logical, the figures are of relatively good quality and the historical background has given credit as is appropriate. The references are adequate. However, the innovation is not clarified adequately. Overall, I think this manuscript can be considered after minor revision if the author could adequately address the comments below.

Specific/Detailed Comments

  1. Page 2: “to evaluate, for the first time, the potential of the RST approach on a time series (2008-20017) of in situ chl-a measurements.” the processing difference or difficulty of the RST approach for in situ data other than the OCR data is needed to clarify better.
  2. Page 2: “Ciancia et al. [30] successfully applied the RST approach to identify anomalous phytoplankton blooms in the Gulf of Taranto (North-Western Ionian Sea) by using 12 years (2003–2015) of MODIS-Aqua Level 3/Level 2 chl-a data.” so what’ the difference of the RST analysis approach compared with previous studies such as Ciancia, just the OCR data and region difference? Need to clarify better for the improvements.

Author Response

Please, see the attached file below.

Reviewer 3 Report

General Comments

The assessment of the RST approach was well-developed and quite strong. Satellite estimates of Chl-a ALICE showed very good agreement with in situ values, showing the potential for applying the satellite product to a spatio-temporal analysis.

The flow and understanding of the paper would be much stronger if Figure 8 and 9 were part of the Results section. As it stands, the results and analysis are quite strong through Figure 6. However, the inclusion of physical oceanographic variables to characterize the convective event appear to be limited to mapping and some temporal analysis of heat flux terms. There is no attempt at spatial-temporal analysis between physical features and high Chl-a ALICE pixels beyond a visual appraisal. It seems all the necessary data is in hand to develop a statistical relationship between physical features of the system and Chl-a ALICE pixel values for this single event in March 2012. Why not perform a cross-correlation analysis to determine a consistent lag between physical features and high Chl-a ALICE pixels? If a lag is established, you could develop a statistical relationship between features within that lag period. That would provide more objective evidence of a relationship and significantly improve the manuscript.

Currently, the Results and Discussion blend together, and the use of the physical data is weaker than it could be. If this is developed more and a clearer distinction between the Results and Discussion is made, this would be a strong, very novel paper. As it stands, there are many, many caveats tying the physical features to anomalous Chl-a values (probably and potential* appear a combined ten times specifically affiliated with Results and Discussion for data/analysis included in Fig. 7-9).

I highly recommend the authors spend more time analyzing these relationships for this single event. Perhaps the data is too complicated to clearly delineate this but that is still a very useful result to show (even if it only appears in the Appendix). Natural systems are noisy – do you have the data needed to draw your conclusions more definitively, or does the community need to acquire additional data to better understand relationships between physical variability and phytoplankton phenology?

At a minimum, the Results and Discussion need a clearer delineation to help with the flow of the paper. This involves moving all presentation of data/analysis (Figs. 8-9) to the Results section and presenting a clearer Discussion of the analyses in totality and relevance compared to the types of analyses that were performed (i.e., visual versus statistical).

Specific Comments

(no lines numbers, so I've done my best to orient the author to the specific passages that require minor edits)

Introduction

Page 1

First paragraph

“[…] thus impacting on fisheries [4-6].”

I recommend re-phrasing to “[…] thus impacting fisheries [4-6].”

Page 2

Second full paragraph

“[…] the major processes driving the nutrients availability into the euphotic layer […]”

I recommend re-phrasing to “[…] the major processes driving nutrient availability in the euphotic layer […]

“In this context, it should be worth […]”

I recommend re-phrasing to “In this context, it is worth […]”

Introduction, final paragraph, second line

(2008-20017) should be (2008-2017)

Materials and Methods

First paragraph

“oligo-trophic” should be “oligotrophic” in this instance

Section 2.5, Second paragraph

Remove the extra “Flux” in “Net Heat Flux Flux”

Section 3.3, first paragraph

“[…] in March 2012 is showed in Figure 6.”

“showed” should be “shown”

Third to last paragraph

“[…] as showed by the sea surface current data (Figs. 7a.d).”

I recommend the following changes:

“[…] as shown by the sea surface current data (Figs. 7a-d).”

Figure 4 is quite difficult to view. I recognize the need to fit this figure on a single page and providing standardized axes; however, is it possible to fit the page/standardization requirements but with larger font (perhaps fewer y-axis labels), larger markers and slightly thicker line width? Perhaps removing repetitive axes labels across common dimensions could also provide more space to make individual panels of the figure larger?

Figure 9a – is there any reason for the inverted colorbar? Typically the bottom of the colorbar is affiliated with the lowest values.

Author Response

Please, see the attached file below.

Reviewer 4 Report

The authors utilize a satellite based chl-a anomaly detection method to the NW Mediterranean. The performance of the method shows good correlation with in-situ data. Posible oceanographic causes for a particular event are presented based on thermal flux data.

Overall, it is an interesting and well written study; concise, well reasoned, and its relevance to the field is explained. I have very few edits to make.

The biggest issue I have is with figure 2 and the conclusions drawn therefrom, where the authors describe the data as a gaussian distribution, when in fact it appears to have a significant positive skew. This should be corrected and commented upon, as the Chl-a ALICE parameter is typically considered more Gaussian and the definition of the anomalous condition is dependent upon the statistics describing the gaussian. This must be addressed and examined for relevance before publication.

Author Response

Please, see the attached file below.

Reviewer 5 Report

I enjoyed reading this manuscript. It is straightforward and to the point without excessive verbosity. The verification of satellite Chl-a products is of significant interest since the experience shows that their suitability varies greatly with location. This study offers a comparison with a long-term dataset of in-situ measurements. I think many readers will find the paper worthy of reading.

Author Response

Please, see the attached file below.

Round 2

Reviewer 3 Report

The authors have adequately addressed prior concerns with revisions.